# Visualization Technology and Deep-Learning for Multilingual Spam Message Detection

Hwabin Lee, Sua Jeong 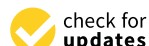, Seogyeong Cho and Eunjung Choi *

Department of Information Security, Seoul Women's University, Nowon-gu, Seoul 01797, Republic of Korea
* Correspondence: chej@swu.ac.kr; Tel.: +82-970-5339

**Abstract:** Spam detection is an essential and unavoidable problem in today's society. Most of the existing studies have used string-based detection methods with models and have been conducted on a single language, especially with English datasets. However, in the current global society, research on languages other than English is needed. String-based spam detection methods perform different preprocessing steps depending on language type due to differences in grammatical characteristics. Therefore, our study proposes a text-processing method and a string-imaging method. The CNN 2D visualization technology used in this paper can be applied to datasets of various languages by processing the data as images, so they can be equally applied to languages other than English. In this study, English and Korean spam data were used. As a result of this study, the string-based detection models of RNN, LSTM, and CNN 1D showed average accuracies of 0.9871, 0.9906, and 0.9912, respectively. On the other hand, the CNN 2D image-based detection model was confirmed to have an average accuracy of 0.9957. Through this study, we present a solution that shows that image-based processing is more effective than string-based processing for string data and that multilingual processing is possible based on the CNN 2D model.

**Keywords:** spam detection; deep learning; visualization technology; CNN

## 1. Introduction

Since the outbreak of COVID-19, reliance on cyberspace has increased, and spam abuse is increasing as well. Smishing, a type of spam, is a combination of SMS and phishing and is a new fraudulent method that steals users' financial information with spam messages containing malicious URLs. However, these smishing instances often show different aspects than previous types of spam. Firstly, there are increases in the rate of smishing, as well as in impersonating public institutions and acquaintances. Secondly, international spam crimes, which are no longer domestic but cross borders, are also emerging. According to the article [1], Federal Trade Commission (FTC), the amount of damage caused by scam text is estimated to be USD 86 million. In addition, the reported number of damages is 334,524 per year, which is equal to 916 cases per day on average. Therefore, spam SMS detection is essential for damage prevention.

To solve this SMS smishing problem, various methods of spam detection have been proposed. In particular, a study [2] classified recent spam detection methods into those using background information (including SMS messages, as well as accounts and behavior features) and those using content-level spam detection (focusing on the content of messages), finding that, among these spam detection methods, text-based spam detection was widely applied. Additionally, in [3], a bag of words (BOW) was used as a text-based content-level spam detection method.

Existing spam detection methods essentially require natural language processing to process input text data. Many researchers have proposed English natural language processing, but the amount of papers with non-English natural language processing is relatively small. In addition, each language's grammatical characteristics make it more complex to process spam content. For example, Korean has linguistic characteristics such as

large differences between spoken and written language; honorifics according to relationships between listeners and speakers; semantic changes according to prosodic elements; frequent omissions of subjects, predicates, and objects; and difficulty in word spacing. However, since there are not many research methods available, it is difficult to process natural languages compared to English. Thus, it inevitably appears differently depending on the natural language processing method and level, as the characteristics are different for each language to detect content level. In a previous study [4], multilingualism showed lower accuracy than English. The researchers proceeded with natural language processing for Spanish, Chinese, and Indonesian [5], but as there were differences in character form and grammatical complexity, they found the same limitations as previous models. In the case of using OCR, spam messages are processed by applying deep learning after natural language processing. In addition, there are difficulties in applying existing detection models to the form of current spam text. Spammers intentionally make typos or alterations, such as creating spaces between words, and preprocessing cannot determine whether these words are spam. In response, ref. [6] used CBOW and skip-gram methods to detect not only general strings, but also texts replaced with special symbols, languages of other countries, and numbers. In this paper, to conduct image-based detection research, spam and ham were classified by applying image-processed SMS to deep learning through separate visualization processing.

In this study, an image-based spam detection method using a CNN 2D model is used to generate Unicode-based images. We conduct research by converting English, which can be considered a basic sample, and Korean, which can be considered a unique sample, into Unicode in our dataset. Our study shows higher accuracy than existing methods. In addition, in another respect, it shows different characteristics from existing spam detection methods using deep learning.

Firstly, the Unicode-based image creation method eliminates preprocessing. Therefore, the comprehensive operation is possible for spam detection not only in English, but also in non-English languages that can be expressed in Unicode or in datasets with mixed languages.

Secondly, we propose an image-based spam detection method that is different from existing general text-based spam detection methods. In this method, the model learns spam and ham dataset images separately and extracts the features of the images to detect spam.

## 2. Related Works

### 2.1. Text Detection and Classification

2.1.1. Character-Based Text Classification

Character-based text classification has been commonly used as a spam detection method. Therefore, many researchers continue to propose spam detection models using machine learning, and model performances have improved over time. The initial model was an RNN model for string classification with the advantage of being able to flexibly create structures as needed, regardless of sequence length. LSTM [7], which is a modified RNN model, has emerged to compensate for the shortcomings of the RNN model for more effective spam classification. As well as LSTM models [8], GRU [9] appeared, but problems still exist with this model. In a later study [10], researchers pointed out the heavy feature engineering of previous models and proposed an improved LGRU model. As such, many researchers have tried to continuously develop detection models by improving and merging existing models or by applying a CNN, which is a representative technology of computer vision systems, in recent years. CNNs [11], which show excellent performances in the NLP field, perform sentence classification with high accuracy. In [12], SVM and naive Bayes machine-learning models showed relatively better accuracy than existing models but could not learn low-level features, which is an advantage of deep learning. In addition, RNNs have the disadvantage that their learning ability deteriorates when processing long input sequences due to the "vanishing gradient" problem [13]. LSTM and GRU mod-

els have appeared to compensate for these shortcomings, but there are still problems. In order to compensate, an SSCL (CNN-LSTM) deep-learning model was used to improve accuracy [14]. The accuracy of each of the above-mentioned models was measured using a dataset from the UCI repository in [15]. In addition, 98.5% accuracy was confirmed for the LSTM model.

The advanced form of a CNN specializes in image classification. Since researchers have found that CNN models can be applied to NLP (natural language processing), they have focused on proposing spam detection models using CNNs. In [4], a CNN was applied to text classification to demonstrate that, despite its structural simplicity, this architecture could produce good effects. However, this method had the characteristic of requiring a preprocessing method and individual preprocessing for each language evaluated. Most of the studies mentioned above have made detection models using English spam datasets. For this reason, many researchers who have conducted NLP using Spanish, Chinese, or Indonesian have not shown as much accuracy as those using English datasets, even if they use the same models, due to a lack of absolute datasets, the complexity of character shapes and grammar, etc. Because of this, in both [16] and [17], it was necessary to perform separate preprocessing according to the characteristics of the language used or to create a library to classify the text. CNNs have obvious limitations, as mentioned. Multilingual datasets can cause inefficient processing and unstable model accuracy.

Globally, spam messages are not limited to English [18]. According to [19], in France, the Netherlands, and Germany, spammers use a spam translation technique to generate spam at 53%, 25%, and 46%, respectively. This has drastically increased the number of spam messages worldwide, and the detection of such multilingual spam requires rules for these languages [20].

For spam detection focusing on filtering, approaches such as the bigram function [21], the average length of a dataset, specific function words and frequencies, and the number of special characters [22] are used mainly for filtering, and language rules are important. In addition, in the case of [23], which used an imbalanced dataset, it was reported that model efficiency was improved with an overfitting reduction using a resampling technique and the preprocessing of many conditions. However, our study did not focus on the filtering process and was not affected by the relationships among words, special symbols, length, and frequency.

### 2.1.2. Image-Based Text Classification

Since the appearance of image-based spam, many researchers have pointed out the necessity of image-based text classification. For image-based spam detection, there are two main types: using an OCR filter and using example-matching approaches.

First, there are spam detection techniques using OCR technology. OCR can largely be divided into two steps: detecting a string in an image and recognizing text. In spam detection using OCR, after conducting text extraction, spam is detected using the text-based spam detection methods described in Section 2.1.1. OCR filtering is widely used, judging from the fact that, the more keywords are found in an extracted text, the higher the likelihood that it is an image, including a spam image. However, this method has a limitation. Attackers complicate the image backgrounds of image-based spam, which prevents OCR filters from detecting character strings so that they cannot be properly recognized [24]. Furthermore, if the keyword list is not updated frequently to compensate for misspellings or deceptions of common words in the character recognition stage, OCR filters have difficulty with image detection [25]. OCR is a technology that recognizes and compares shapes rather than interpreting characters, and OCR-based technology is computationally expensive.

Second, there are example-matching approaches. These detect spam by using low-level features, such as image color information. These features measure an image's similarity to test images and determine whether it is spam based on the results, which are also different, even if an attacker introduces a slight change (font, text size, typo, etc.) to a

spam-based image. Subsequently, CNN methods can be used to detect images with very high classification accuracy [26,27].

Image spam is a technique that has emerged to evade text-based spam filters. In [27], images marked with pure text on blank images were used as the image spam dataset. These images were composed of text, such as product advertisements and inducements to collect personal information. These images looked like text emails to the casual observer, but they were actually images. To detect image spam with high accuracy, we applied a novel feature set consisting of a combination of raw images and Canny edges based on a CNN 2D model. The experiment used an SVM, an MLP, and a CNN. Among them, CNN showed the highest accuracy in the challenging image-spam-like dataset.

The research in [28] presented a new approach to NLP tasks. In existing text string classification methods, progress has always been based on text, but in this study, images were used. The results showed that both Chinese text classification tasks and English text classification tasks were performed with high accuracy using a CNN 2D model by receiving images as input. This method demonstrated that semantic features could be obtained from images with text without optical character recognition (OCR) or natural language processing. The aim of our research was to propose an efficient method without natural language processing. Through this, our study also attempted to suggest an efficient method that did not process natural language.

### 2.2. Spam Detection Using ML and DL

### 2.2.1. Utilization of Machine Learning

The application of machine learning [29] used two datasets and eight classifiers to perform spam detection. The classifiers used in this study were largely classified into two types: traditional machine learning and deep learning. The detection results according to the classifier were evaluated and compared based on accuracy, precision, recall, and the CAP curve. The CNN classifier, a deep-learning method, showed the highest results, with values of 0.9926 and 0.9994 for the datasets. CNNs are commonly used for image data classification, but they also showed the highest results for textual data. Among the traditional classifiers, SVM and NB showed good results and were closest to the CNN for both datasets.

The research in [12] comprehensively reviewed spam email detection studies using machine learning and summarized the characteristics of spam email detection studies. First, certain algorithms (NB and SVM) had high demand compared to other machine-learning algorithms. Second, most systems used a combination of algorithms, not a single ML algorithm, for higher accuracy. Third, most studies focused on BOW and body text as the features of e-mails and proposed future research opportunities that focused more on other features [30].

TF-IDF (term frequency–inverse document frequency) and Random Forest were used to collect spam SMS message data in this experiment [31]. TF-IDF was used in the feature extraction step. TF-IDF evaluates importance according to the number of occurrences of a word in a document and applies a weight. In the classification stage, various algorithms, such as MNB, KNN, SVM, DT, and Random Forest, were used. The TF-IDF and RF combination outperformed the others in terms of accuracy percentage.

### 2.2.2. Utilization of Deep Learning

As with machine learning, extracting and representing the features of data are essential, but these are difficult and have limitations when applied on a large scale. However, deep-learning technology is capable of extracting higher-level features from a large number of lower-level features. It performs better on difficult tasks with large amounts of data than traditional machine-learning methods [32].

Deep learning is considered to be the most suitable method for spam detection, and many related studies have been proposed. The researchers in [33] used an RNN model for spam detection. It was compared with SVM and naive Bayes models, which are widely

used for spam detection. In the proposed method, the preprocessing and feature extraction steps were performed as part of the classification process, which showed better results than existing machine-learning models. In this paper, machine-learning models, such as SVM and naive Bayes models, showed relatively good accuracy, but low-level feature learning, which is an advantage of deep learning, was not possible.

Regarding the limitations of this study, this paper focused on an SMS spam detection model that could detect well regardless of the linguistic specificity of individual languages using a CNN 2D model.

Table 1 provides a summary of classification research presented in Section 2.

**Table 1.** Summary of related works.

| Classification | Reference | Keywords |
|---|---|---|
| Character-based text classification | [7–23] | Spam Detection, SMS Data, Twitter, Sentence Classification, Support Vector Machine (SVM), Machine Learning, Feature Extraction, Text Mining, Sentiment Analysis, Filtering, Recursive Neural Network (RNN), Long Short-Term Memory (LSTM), Convolutional Neural Network (CNN), Natural Language Processing (NLP) |
| | [18,19] | Spam Trend |
| Image-based text classification | [24–28] | Image Spam, Visual Bag of Words, Spam Filtering, Spam Detection, Convolutional Neural Network (CNN), Social Media, Support Vector Machine (SVM), Image Spam, Document Categorization, Multilayer Perceptron |
| Spam detection using ML/DL | [29] | Machine Learning, Sentence Classification, Support Vector Machine (SVM), Naive Bayes (NB) |
| | [30,31] | TF-IDF, Random Forest, Bag of Words (BOW), ADASYN |
| | [32,33] | Deep Learning, Time Series Data, SMS Data |

## 3. Proposed Method

The proposed spam detection framework is shown in Figure 1 and consisted of three processes: preprocessing (NLP, preprocessing, and visualization), training, and classification. Through the process of preprocessing, various visualizations of spam and ham messages were performed. The visualization proposed two methods: grayscale and RGB. This is explained in detail in the next section. In this paper, learning was conducted for various models, including RNN, LSTM, and CNN 1D models, but the method proposed as the optimal model among them achieved learning using a CNN 2D model. In the last step, spam and ham classification was performed through a dense layer using sigmoid activation. A description of each process follows.

### 3.1. Datasets

Three types of datasets were used: English, Korean, and a mixed dataset (English and Korean). For the English dataset, the "SMS spam collection dataset" published on Kaggle was used, and for the Korean dataset, we visited the Cyber Security Big Dataset Center and used a Korean smishing dataset in addition to collecting additional data from internet crawling and the open dataset "One-shot conversation dataset with Korean emotional information" from AI Hub. There were three types of datasets used in this study: an English ham and spam dataset, a Korean ham and spam dataset, and a mixed Korean and English dataset. The datasets contained 5568, 69,654, and 75,222 samples, respectively. Spam data were labeled as 1, and ham data were labeled as 0. Table 2 shows the summary of the datasets.

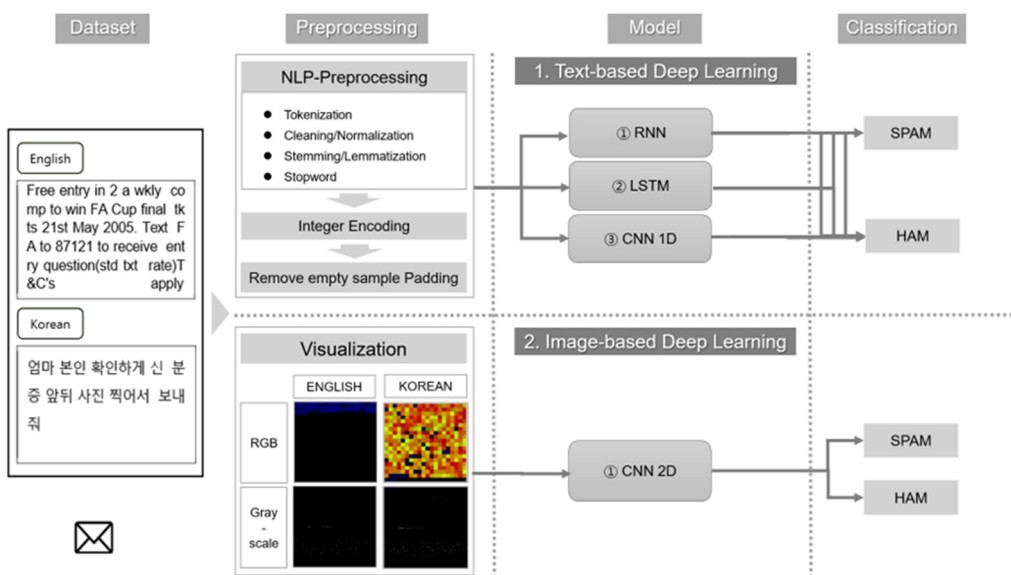

**Figure 1.** General description of proposed spam detection methods.

**Table 2.** Dataset matrix.

|  | Spam | Ham | Total |
|---|---|---|---|
| English | 743 | 4825 | 5568 |
| Korean | 30,674 | 38,980 | 69,654 |
| English and Korean | 31,417 | 43,805 | 75,222 |

### 3.2. Application of Text-Based Deep Learning

In this paper, research was conducted with RNN, LSTM, and CNN 1D models, which have been suggested by many researchers as spam and ham detection models, to compare our accuracy results with those of existing studies.

#### 3.2.1. NLP Preprocessing

The process by which a model receives natural language as input is not simply a process of receiving a sentence literally. In general, a model transforms characters into numbers through tokenization, word set generation, integer encoding, padding, and vectorization and then proceeds with learning. In this process, strings are greatly affected by language characteristics. Therefore, in this paper, we conducted practice by adding a preprocessing step that considered language characteristics, as well as basic preprocessing steps for each Korean and English dataset.

#### Tokenization

Tokenization is the step in which data are cut into words or characters. In the case of the English dataset, we proceeded with tokenization in two ways. First, we distinguished upper- and lowercase letters and converted them uniformly into lowercase letters. Second, we removed stop words using the "English" stop word library from NLTK. In the case of the Korean dataset, there were difficulties in tokenizing according to spaces, as Korean has a variable word form, which changes form depending on the context, even if it is the same word. Therefore, instead of spacing, we used word normalization, and morpheme tokenization was performed using a morpheme analyzer. In addition, as a lot of SMS data include special symbols, characters, etc., so a process of blanking out all characters except those in English and Korean was added.

Integer Encoding

In this step, the data were treated as numbers. It could largely be divided into the process of creating a word set and the process of assigning an index to each word set. First, both Korean and English data were tokenized, and each word was given a unique integer. Furthermore, when the data were processed, it was determined that data with significant effects on classification would appear at least twice, so data with frequencies of less than two were removed.

Removing Empty Samples and Padding

In this step, after conducting the previous two steps, empty data were deleted and padding was performed to make data of different lengths equal.

3.2.2. Models

1. RNN

$$h_t = \tan h(x_t W_{ih}^T + b_{ih} + h_{t-1} \ W_{hh}^T + b_{hh}) \tag{1}$$

Ref. [34] RNN is the most basic form of a sequence model that processes inputs and outputs in sequence units and is a model that mainly deals with sequential data or time series data. These deep-learning algorithms classify the next input based on the information obtained from the previous input and show excellent performances in use for continuously correlated data, such as language translation, natural language processing (NLP), and voice recognition. In this paper, the model was designed with the parameter values shown in the figure below, and a simple RNN was used as the model. Table 3 shows the settings of RNN model

**Table 3.** Settings of RNN model.

|  | Optimizer | Epochs | Batch_Size | Validation_Split |
|---|---|---|---|---|
| RNN | RMSprop | 20 | 64 | 0.3 |

2. LSTM

$$i_t = \sigma(W_{ii}x_t + b_{ii} + W_{hi}h_{t-1} + b_{hi})$$

$$f_t = \sigma(W_{if}x_t + b_{if} + W_{hg}h_{t-1} + b_{hf})$$

$$g_t = \tan h(W_{ig}x_t + b_{ig} + W_{hg}h_{t-1} + b_{hg})$$

$$O_t = \sigma(W_{io}x_t + b_{io} + W_{ho}h_{t-1} + b_{ho})$$

$$C_t = f_t \odot C_{t-1} + i_t \odot g_t$$

$$h_t = o_t \odot \tanh(c_t) \tag{2}$$

Ref. [35] An LSTM model is also a type of sequence model that can proceed with interactions among cell states without significantly changing the overall information. We designed the model in this paper with seven layers, including a dropout layer, a batch normalization layer, a dense layer, and so on. In the first layer, the input came as an embedding layer, passed through the next two LSTM layers, and dropped some nodes while passing through the dropout layer. Afterward, the batch_normalization layer went through the

process of stabilizing the learning process as a whole and passed through three dense layers. At this stage, the last layer, Dense_1, output one final value by distinguishing whether the final value the model wanted to predict was spam (1) or ham (0), so the activation function was designated as sigmoid and dense (1) at the same time. Table 4 shows the settings of LSTM model. Figure 2 shows the structure of LSTML model.

**Table 4.** Settings of LSTM model.

|  | Optimizer | Epochs | Batch_Size | Validation_Split |
|---|---|---|---|---|
| LSTM | adam | 100 | 64 | 0.3 |

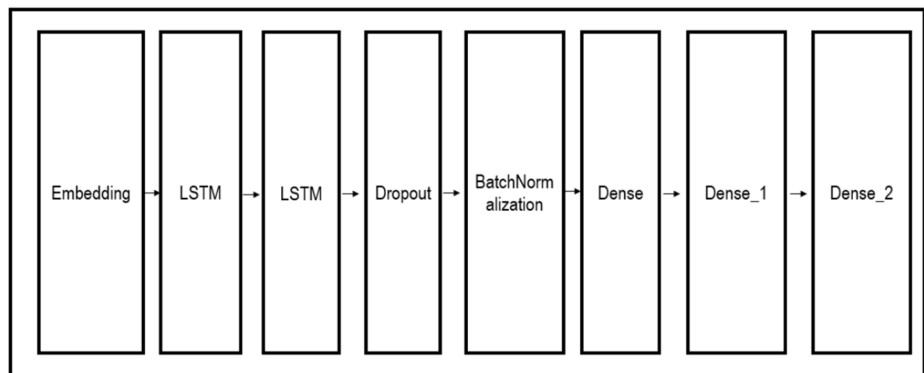

**Figure 2.** Structure of LSTM model.

3. CNN 1D

$$\text{out}\left(N_j, C_{out_j}\right) = \text{bias}\left(C_{out_j}\right) + \sum_{k=0}^{C_{in}-1} \text{weight}\left(C_{out_j}, k\right) * \text{input}(N_i, k) \tag{3}$$

Ref. [36] CNN 1D is a convolutional neural network model (CNN). A CNN's basic concept is to look at data as pictures. There is no exception for strings. A CNN 1D model receives a vector as input that is created by stacking word embeddings of sentences and processes it by performing 1D filtering from top to bottom. In addition, the model is frequently referred to as the most-used model in the design of natural language processing (NLP). In this paper, the model was constructed as follows: overall, the model was similar to the configuration of the LSTM model mentioned above, and a global_max_pooling layer was added to prevent overfitting. Figure 3 shows the Text-based CNN 1D preprocessing. Table 5 shows the settings of CNN 1D model.

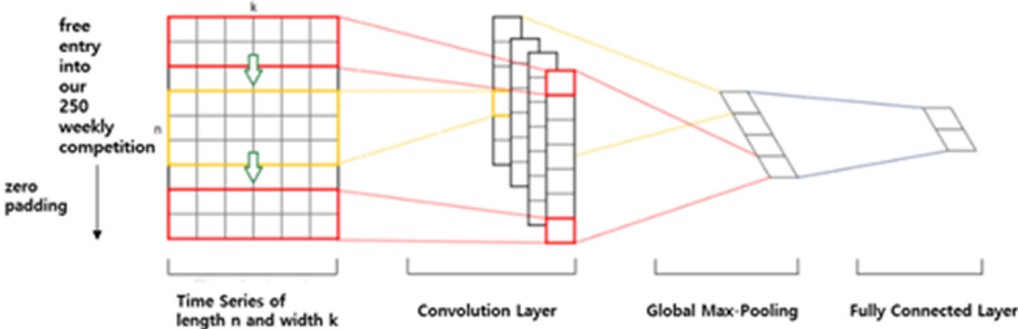

**Figure 3.** Text-based CNN 1D preprocessing.

**Table 5.** Settings of CNN 1D model.

|        | Optimizer | Epochs | Batch_Size | Validation_Split |
|--------|-----------|--------|------------|------------------|
| CNN 1D | adam      | 100    | 32         | 0.3              |

### 3.2.3. Experiment Results

When using each model, the accuracy was as follows: overall, it was confirmed that the detection accuracy for Korean was higher than that for English by about 0.01 percentage point. As a result of the experiment, the LSTM model showed the highest accuracy for the mixed dataset. Table 6 shows the performance of each model accuracy per dataset.

**Table 6.** Performance of each model accuracy per dataset.

|        | English | Korean | English and Korean |
|--------|---------|--------|--------------------|
| RNN    | 0.9815  | 0.9914 | 0.9846             |
| LSTM   | 0.9843  | 0.9929 | 0.9947             |
| CNN 1D | 0.9843  | 0.9955 | 0.9938             |

### 3.3. Application of Image-Based Deep Learning

Figure 4 shows general decription of spam detection using image-based deep learning.

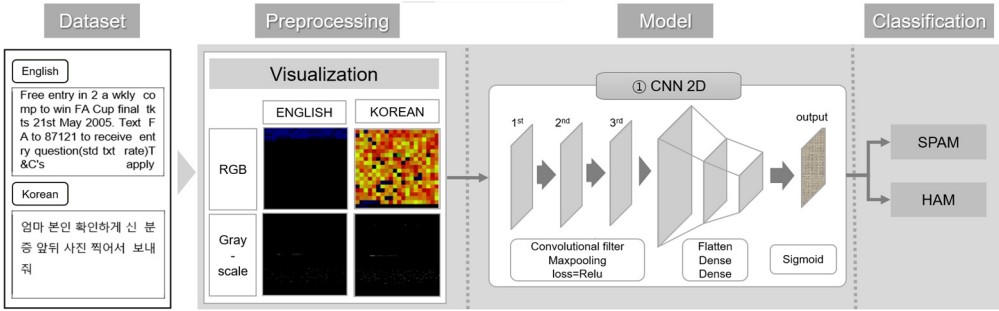

**Figure 4.** General description of spam detection using image-based deep learning.

### 3.3.1. Preprocessing

Both the grayscale and RGB image-processing methods involved converting characters to Unicode values. In this process, blank characters in the input string were removed. This was because blank characters are considered to be a linguistic characteristic, as the presence or absence of blank characters is different for each language, and meaning is created through blank characters. Therefore, blank characters were removed to exclude language characteristics as much as possible. On the other hand, special characters were not removed because they were considered part of the preprocessing process, which had nothing to do with the grammatical characteristics of the languages. Table 7 shows unicode and coordinate values of characters. Figure 5 shows the Grayscle images of 'A', '가'. Figure 6 shows the RGB images of 'A', '가'.

**Table 7.** Unicode and coordinate values of characters.

| String | Unicode         | Grayscale | RBG       |
|--------|-----------------|-----------|-----------|
| A      | $0 \times 0041$ | (0,65)    | (0,0,65)  |
| 가     | $0 \times AC00$ | (172,0)   | (172,0,0) |

'가' is the first syllable in the standard ordering of Hangul, the Korean alphabet.

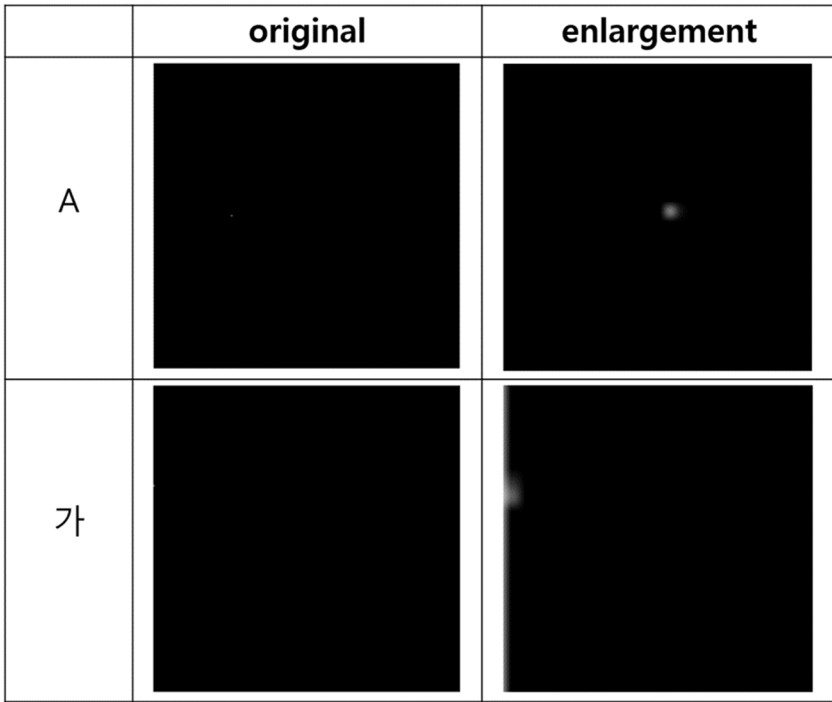

**Figure 5.** Grayscale, 'A', '가'.

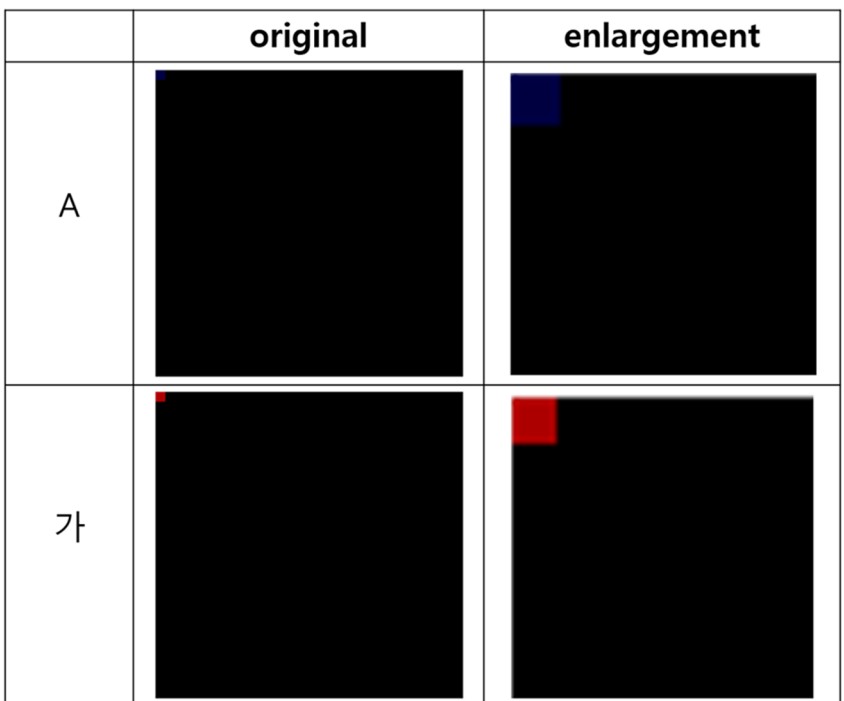

**Figure 6.** RGB, 'A', '가'.

1. Grayscale Image

In this step, the Unicode of the character was converted into a pair of coordinates (X,Y) for conversion into a grayscale image. When text was entered, the Unicode value was extracted from each character. The extracted value was divided by one byte and matched with (X,Y) coordinates. 256∗256 was used to increase the value of an element corresponding to each coordinate in the image. The values of the elements in the array contained the pixel brightness values. The brightness of one pixel ranged from 0 to 255. For example,

a character whose Unicode value was 0xff00 corresponded to the coordinates (255,0) and increased the value of the corresponding element.

When encoding with Unicode, English is 1 byte per character, and Korean is 2 bytes per character. Since an English character is 1 byte, the x-value is always 0 when converting to a coordinate. The coordinate values of English characters are all located on the edge of an image, making it difficult to identify them. To solve this problem, instead of using 0 for x, we used 128, which was the median of the coordinate plane. Since the 128 (+UC80XX) area was the value of the CJK (China, Japan integrated Chinese and Korea) character area, there were no characters that overlapped for the Korean, English, or Unicode special symbols. Figure 7 shows changing process from text to a grayscale image

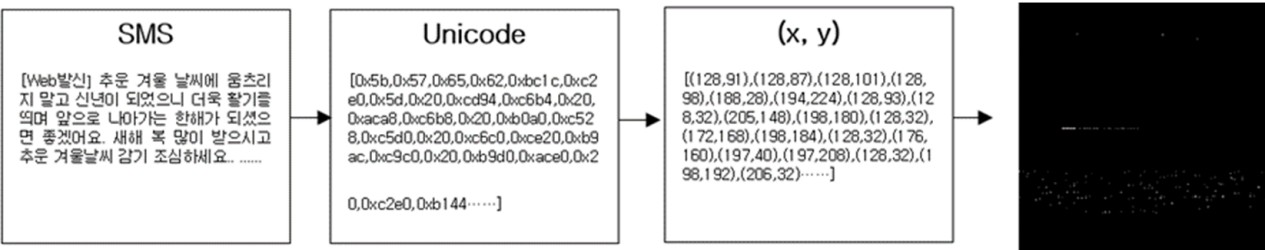

**Figure 7.** Changing process from text to a grayscale image.

256∗256 pixel values were initialized to zero, and the Unicode value of each character was mapped as a coordinate (X,Y). In addition, if the length of a string was short, the mapped coordinate values were very small, making them difficult to identify. To solve this problem, when the values of the array elements were taken at once, the brightness per pixel was increased by increasing the value by 200 so that it could be seen more clearly. Through this process, a grayscale image was finally generated.

2. RGB Image

The RGB visualization method creates an image by converting the Unicode of a character to a value corresponding to its RGB position and fills pixels with a color in order to obtain a 31∗31-sized image initialized to 0. Three-dimensional data, which are RGB three-pixels expressed in real numbers per pixel, are used for color image production. If N filters are applied to the convolution layer, the output data have N channels. In this study, three channels were classified by language using RGB, and since multiple languages were used, colors were applied according to the number of bytes. Figure 8 shows CNN for RGB images.

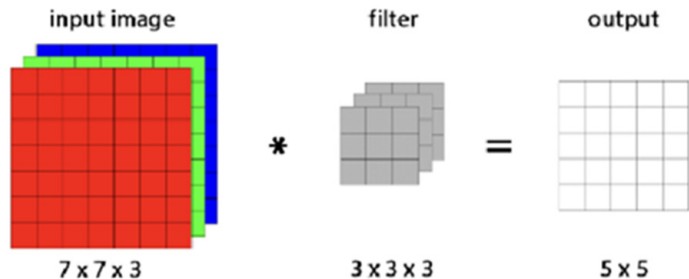

**Figure 8.** CNN for RGB images.

This model changed each character to a Unicode value to distinguish languages with different bytes when creating RGB images. The changed character was secondarily converted to a decimal to match the size of 0-255 for the image to be displayed in the two models. Korean, which was 2 bytes, was set to RG, and English, which was 1 byte, was set to B, while the other channels were assigned as 0. Mixed languages were sometimes

used in the dataset, so colors were assigned to avoid overlapping. Among the 2-byte languages, the Korean language used by our model had over 10 times more types than English Unicode. In order to prevent pixels that were concentrated on one side or that were not captured in certain parts, an image was created using RGB. Through this process, a tuple for the sentence at the bottom was created, and the image shown in Figure 9 was output. The size of the output image of the convolution layer contributing to the output of all channels of the input was ( nOutputPlane × height × width ) (value varied according to batch size).

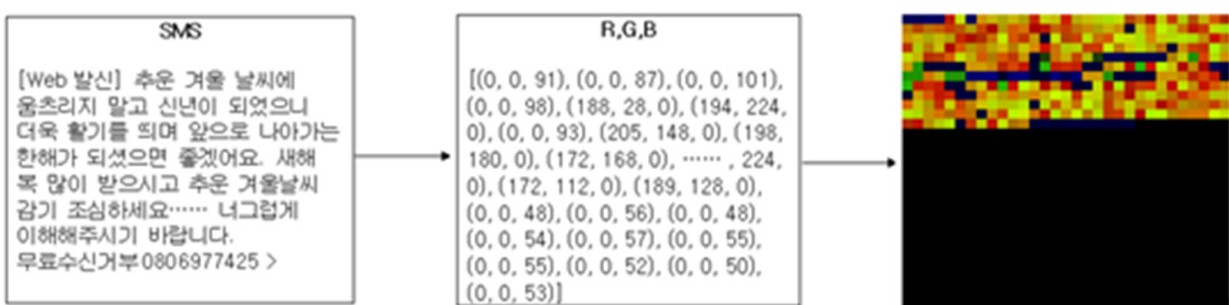

**Figure 9.** Each character was converted to one pixel of an image.

Even if only two channels were used out of the total three channels, six separate filters must be operated because each output channel was affected by all the input channels. The output value could be calculated as the sum of the input channels and convolution filters. Figure 9 shows the process of converting each character into one pixel of an image.

### 3.3.2. Models

Figure 10 provides illustration of the architecture of CNN 2D processing image data. To classify the visualized spam and ham data, a standard CNN was used. Deep learning is an advanced form of the model in the neural network field of artificial intelligence, and their hidden layers consist of several stages.

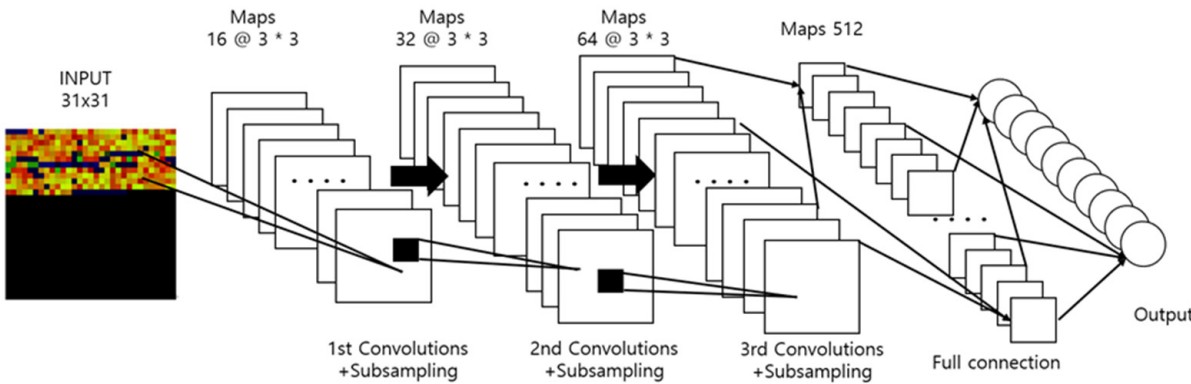

**Figure 10.** Illustration of the architecture of CNN 2D processing image data.

Convolutional neural networks (CNNs) mimic biological visual processing and are the most commonly used artificial neural network for visual image analysis and classification. By not combining all nodes, the method is used to reduce the amount of computation and increase efficiency. By maintaining the spatial information of an image, it effectively recognizes and emphasizes data as features with adjacent images. CNN models sconsist of parts that extract and classify the features of an image.

A convolutional layer extracts visual features of images, such as lines and colors. In this case, image extraction was performed effectively using a filter to minimize the number of shared parameters. A fully connected layer classified images based on the discovered features. All nodes with the result data processed thus far were connected and presented as a one-dimensional array.

Looking at Figure 11, the local receptive field, which refers to the lower square in the input, is connected to the node and filter (weight) of the next layer. In this case, the input data size of the feature map should be the same. Applying a filter to all areas of the input and passing the node of the panel is called "convolution." At this time, there are n different filters, and these filters can recognize one feature.

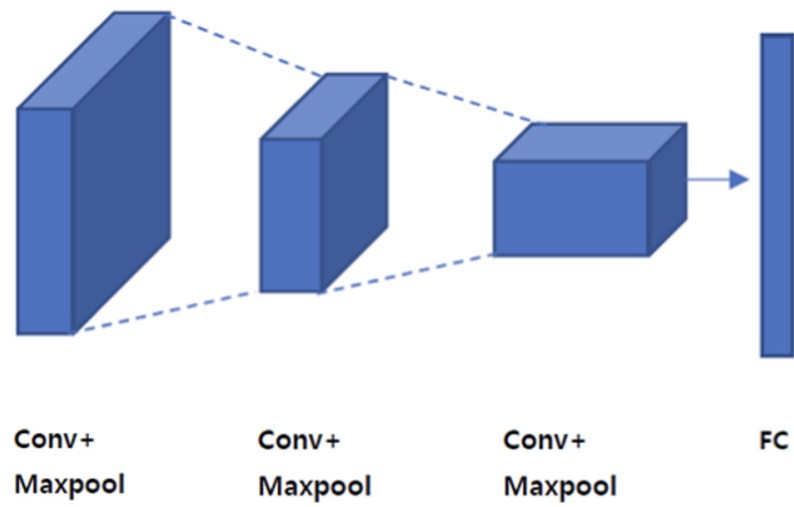

**Figure 11.** CNN structure.

The CNN structure included three convolutional layers (one convolutional operation and one max-pooling operation), a fully connected layer, and an output layer. This model was constructed by repeating the convolutional layer Conv2D and the max-pooling layer MaxPooling2D six times, as well as overlapping the dense layer twice. (Conv2D(a,b,c), padding, input_shape = (d,e,f), activation) was the configuration for the CONV2D layer that extracted features with filters. In this case, the first argument was the number of filters, and the second argument was the kernel (row, column). Padding defined the boundary-processing method, and the input shape was the input shape excluding the number of samples as (row, column, channel); activation referred to setting the activation function. A filter indicated a weight, which could be expressed as a dense layer that determined the number of weights. The first factor values of Conv2D (a,b,c) were increased to 16, 32, and 64, respectively, while the second-factor value remained constant at (3,3). Activation used the rectifier function of "relu." Since the input shape was defined only in the first layer, it was fixed as (256,256,3). Dense applied 512 output neurons to the activation function "relu" and 1 output neuron to the sigmoid. The max-pooling layer extracted the main values of the output image so that it was not affected by minor changes when producing a small-sized output image. This model repeated layer 3 and set the pool size to (2,2) to reduce the vertical–horizontal size by half. The CONV neural network model specified target_size to set the patch image size and batch_size as 32 and (150,150), respectively. Classification class _model set the binary to return a 1D binary label. In all the classification experiments, the learning rates were set to 0.001. All filters in the convolutional layers used 3∗3 filters. Each layer used 16, 32, or 64 filters. A fully connected layer consisted of 512 units. In the final step, for binary classification, the activation function of the sigmoid with one output neuron was used. Figure 12 provides an illustration showing our image-based spam detection process. Table 8 shows the settings of CNN 2D model.

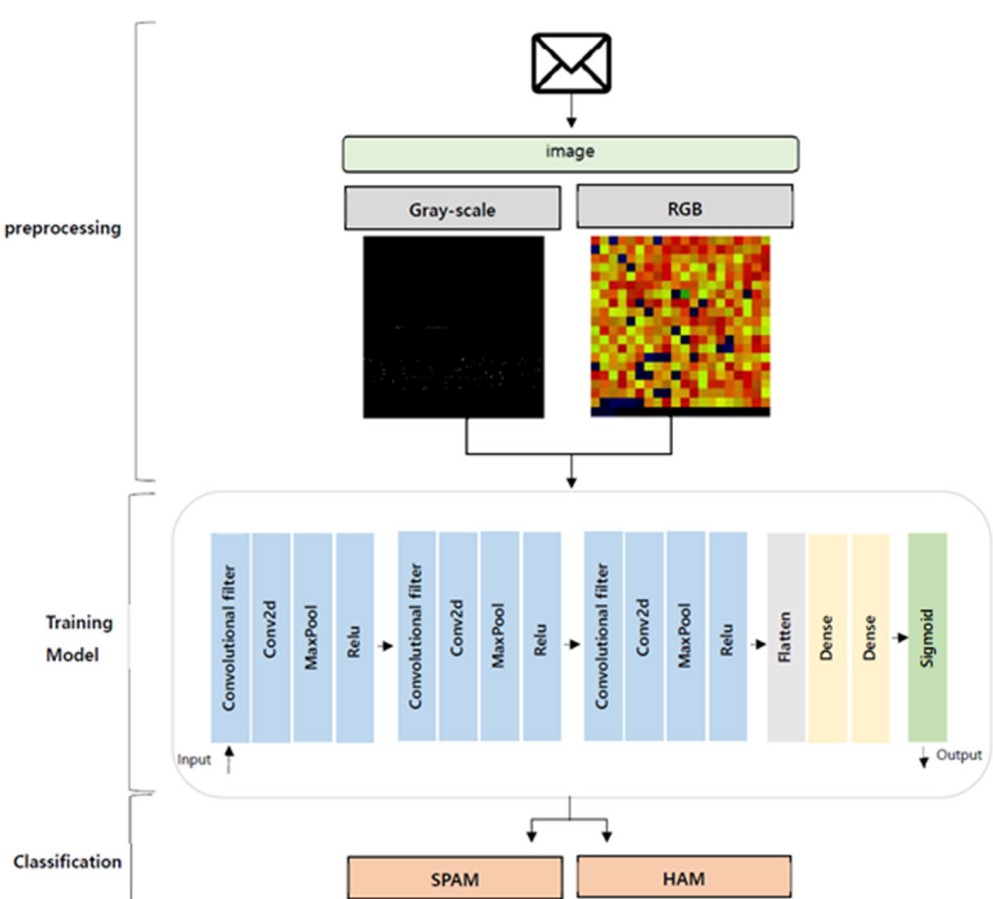

**Figure 12.** An illustration showing our image-based spam detection process.

**Table 8.** Settings of CNN 2D model.

| CNN 2D | Optimizer | Epochs | Batch_Size | Validation_Split |
|--------|-----------|--------|------------|------------------|
| Grayscale | RMSprop | 100 | 32 | 0.3 |
| RGB | RMSprop | 100 | 32 | 0.3 |

### 3.3.3. Experiment Results

In this section, the effectiveness of the multilingual spam classifier is demonstrated by applying the visualization method proposed above to the experimental data. The experiment shows that meaningful classification was possible with visualization using three types of data sets: English, Korean, and an English–Korean mixture. The two visualization methods proposed earlier were compared with the accuracy of each dataset. Table 9 shows spam detection performance using CNN 2D images.

**Table 9.** Spam detection performance using CNN 2D images.

| Method | English | Korean | English and Korean |
|--------|---------|--------|--------------------|
| Grayscale | 0.9967 | 0.9838 | 0.9903 |
| RGB | 0.9981 | 0.9995 | 0.9998 |

When the grayscale visualization method was applied, English achieved an accuracy of 99.67%, Korean achieved 98.38%, and Mixed achieved 99.03%. In the case of RGB, 99.81%, 99.95%, and 99.98% classification accuracies were achieved for the cases of English, Korean, and Mixed, respectively. Both the grayscale and RGB error rates of verification and testing did not differ by more than 0.1%.

In addition, when evaluating the overall performance, the RGB dataset showed a slightly higher recall than the grayscale dataset. The accuracy of smishing detection through text visualization could be determined based on the results when a conversion resulted in an RGB image showing a higher performance than a grayscale image.

## 4. Discussion

Our work has potential for improvement. First, the UCI spam data used as the English dataset included about 5000 samples. Since more than 10,000 Korean datasets were used, there was a limitation in that the dataset was not balanced. Thus, the experiment could be conducted by adding more English spam data. Second, in addition to the spam SMS classification task, an image-processing method could be used for applications in more fields. Third, in this experiment, a standard CNN was used for image-based classification, but variations of CNN models could be applied. Fourth, in addition to Korean and English, other languages could also be applied. Finally, although this study used an existing model, further research should be expected to compare detection using a PLM model, one of the newly proposed detection models, or to study the efficiency of combinations with CNN models.

## 5. Conclusions

Existing spam detection methods are based on strings, and classification is performed through machine-learning or deep-learning models through natural language processing. In this paper, a visualization step was added to an existing detection process for effective spam SMS detection.

The visualization method was largely divided into two types. The first method converted each character's Unicode value into a pair of coordinates. The second method stored each character's Unicode value in an RGB tuple and returned it as a pixel in the image.

A CNN was used as a deep-learning model for spam SMS detection. As a result of applying the deep-learning model after visualization, it was confirmed that there was no difference from the results of the text-based classification experiment. The preprocessing step was eliminated through image processing, and the classification results were outperformed. The proposed visualization method could perform more efficient classification tasks by omitting preprocessing tasks, such as tokenization, stemming, and spell-checking.

**Author Contributions:** Conceptualization, methodology, visualization, validation, software, formal analysis, writing-original draft, writing review and editing, data acquisition, investigation—H.L.; Conceptualization, methodology, Investigation, visualization, validation, software, writing-original draft, writing review and editing, data acquisition, resources—S.J.; Conceptualization, methodology, Investigation, visualization, software, writing-original draft, writing review and editing, data acquisition, data curation—S.C.; funding acquisition, Project administration, supervision —E.C. All authors have read and agreed to the published version of the manuscript.

**Funding:** This research was funded by [Seoul Women's University] grant number [2022-0164, 2022-0214].

**Data Availability Statement:** The English data that support the findings of this study are available in UCI ML repository at Kaggle. These data were derived from the following resources available in the public domain: [https://www.kaggle.com/datasets/uciml/sms-spam-collection-dataset]. The "One-shot conversation dataset with Korean emotional information" dataset are available in from AI Hub. These data were derived from the following resources available in the public domain: [https://aihub.or.kr/aihubdata/data/view.do?currMenu=115&topMenu=100&aihubDataSe=realm&dataSetSn=86]. The Korean smishing data that support the findings of this study are available on request from Cyber Security Big Dataset Center. The data are not publicly available due to their containing information that could compromise the privacy of research participants.

**Conflicts of Interest:** The authors declare no conflict of interest.

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
