# Peer review of "Visualization Technology and Deep-Learning for Multilingual Spam Message Detection"

_electronics, doi:10.3390/electronics12030582_

Round 1

Reviewer 1 Report

Paper title: Visualization technology and deep learning research for multilingual spam message detection

 There are some points that need to be further clarified:

1-       The motivation for the study should be further emphasized, particularly; the main advantages of the results in the paper comparing with others should be clearly demonstrated. 

2-       The limitations of the study are better suited for the discussion in a separate sub-section after the discussion on results.

3-       The example section needs to be further expanded and include some remarks to show the effectiveness and efficiency of the proposed method, compared with others. 

4-       Some remarks on the main results would be necessary and helpful. 

5-       The literature review should be extended.

Recommendation: According to all these issues,

Decision: major revisions

Author Response

Dear , Reviewer 1,

Below is the answer to 'Comments and Suggestions for Authors', which you suggested and asked about our paper. And  For English writing, we assigned MDPI English editing service and got consulting on our overall style of sentence and vocabulary. Thank you for your nice comments. Please see the attachment.

Point 1: The motivation for the study should be further emphasized, particularly; the main advantages of the results in the paper comparing with others should be clearly demonstrated

Response 1: Thank you for your suggestion. We emphasized our research’s motivation and contribution points by adding explanations in (p.2-3 , lines 69-82), introduction section.  

Point 2: The limitations of the study are better suited for the discussion in a separate sub-section after the discussion on results.

Response 2: Thank you for your suggestion. We added a separate section ‘5. Discussion’ (p.16-17, lines 472-483), and shifted explanation about our research’s limitation in this section.

Point 3: The example section needs to be further expanded and include some remarks to show the effectiveness and efficiency of the proposed method, compared with others.

Response 3: Thank you for your suggestion. Same as Point 1, we expanded about our effectiveness and efficiency  in  (p.2-3 , lines 69-82) , introduction section

Point 4: Some remarks on the main results would be necessary and helpful.

Response 4: Thanks for the suggestion. We created Table 9 in (p.16, lines 446) to more accurately state the results. 

Point 5: The literature review should be extended. 

Response 5: We have added the literature review, related text detection[20,21,22] (p.4, lines 128-133), deep learning [30] (p.5, lines 191-193) and NLP [4](p.2, lines 56-58).

Thank you.

Reviewer 2 Report

Dear Authors, You have done good work but need improvement in article structure to improve the quality.

-- Add contribution with highlights.

-- Add related work summary in table.

-- Consider the following paper for citation and related work.

Spam SMS filtering based on text features and supervised machine learning techniques

Detection of fake job postings by utilizing machine learning and natural language processing approaches

Comparative analysis of machine learning methods to detect fake news in an Urdu language corpus

Electroencephalogram Signals for Detecting Confused Students in Online Education Platforms with Probability-Based Features

Building Heating and Cooling Load Prediction Using Ensemble Machine Learning Model

Mez: An adaptive messaging system for latency-sensitive multi-camera machine vision at the iot edge

Mez: A Messaging System for Latency-Sensitive Multi-Camera Machine Vision at the IoT Edge

A Novel Application/Infrastructure Co-design Approach for Real-time Edge Video Analytics

-- When you combine deep learning models, did you consider complexity as a tradeoff between accuracy and efficiency?

-- You combine models using any voting criteria?

-- If figures 1 and 2 are representing the same information you can remove 2. The color scheme is not good. difficult to read.

-- Mention computational time with proposed approach in comparison with other studies.

Author Response

Dear , Reviewer 2,

Below is the answer to 'Comments and Suggestions for Authors', which you suggested and asked about our paper. And  For English writing, we assigned MDPI English editing service and got consulting on our overall style of sentence and vocabulary. Thank you for your nice comments. Please see the attachment.

Point 1:  Add contribution with highlights.

Response 1:Thank you for your suggestion. We emphasized our research’s contribution points by adding explanations in (p.2-3 , lines 69-82), introduction section. 

Point 2: Add related work summary in table.

Response 2: Thank you for your suggestion. We added a new Table 1 in (p. 5, lines 217) .

Point 3: Consider the following paper for citation and related work.

Response 3: 

Thank you for your suggestion. Among the proposed paper, (

  • Spam SMS filtering based on text features and supervised machine learning techniques,
  • Detection of fake job postings by utilizing machine learning and natural language processing approaches,
  • Comparative analysis of machine learning methods to detect fake news in an Urdu language corpus

) , which is most relevant to our study , [22] in (p.4, line 131-133), [30] in (p.5, lines191-193),  [4] in (p.2, 56-58) . In addition,  [20] in (p.4,line 128-129) ,[21] in (p.4, line 129-131) added related works.

Point 4: When you combine deep learning models, did you consider complexity as a tradeoff between accuracy and efficiency?

Response 4:You have raised an important point, however, we believe that it would be outside the scope of our paper. Because our model was not combined. We used models, which were single type models.

Point 5: You combine models using any voting criteria?

Response 5: Same as response 4, we did not combine the models. Each model was run independently. 

Point 6: The limitations of the study are better suited for the discussion in a separate sub-section after the discussion on results. If figures 1 and 2 are representing the same information you can remove 2. The color scheme is not good. difficult to read.

Response 6: We agree with the reviewer’s suggestion. We have revised the manuscript accordingly. 

Point 7: Mention computational time with proposed approach in comparison with other studies.

Response 7: Thank you for your suggestion. But our experimental environment was a high performance computer environment. And We did not check the time because there was not much time delay. The environment in which the experiment was conducted used a dataset that could not be leaked from the special closed network of the center where access permission took a long time. Because of these problems, it is difficult to conduct the experiment again in the near future under the current situation. So, in the current situation, it is difficult to describe the computational time. 

Thank you.

Reviewer 3 Report

The paper deals with SMS spam detection task with deep learning technology. It insists the importance of capability of multilingual SMS spam detection and suggests a new method that first converts each SMS text into an image of fixed size using Unicode value of each character in text and then pass the image into 2D CNN for training. This visualization idea for SMS spam detection seems to be a novel method, but there are too many unclear, incomplete, unnecessary, or missing-the-point expressions in the sentences and the section organization should be reorganized to clarify and focus on the core idea of the paper by reducing or removing too common descriptions.

First of all, the English writing should be much improved by consulting with professional English editors.

The citation reference position and style also should be corrected and improved. For example, “[1] According to …” in line 37-38, “Because of [14] and [15], “ in line 113, … should be clarified and corrected.  

Section 2 (related works) consists of two subsections: ‘text detection and classification’ and ‘visualization detection’. The section subsection is expected to describe existing works on conversion of text (string) into image, but any referred works do not deal with such conversion. I cannot distinguish the referred works in the second subsection from the referred works in the first subsection. And the referred works in the first subsection are also not well-classified and well-described. Similar(duplicate) description is repeated with different related works (1st and 2nd paragraphs in 2.1.1).

The authors should classify and describe recent existing works related to the suggested approach in the target task with which points are similar and which other points are different or unique. Most recent works in text mining field use pre-trained Language Model(PLM) like BERT and fine-tune it some output layers on top of PLM for each down-stream task like spam detection. The related works do not describe or compare the existing works on PLM-based spam detection to the suggested approach. 

The section 3 (proposed method) describes two different approaches with datasets. However, the first approach (in 3.2) must not be the suggested one by the authors. Instead, it would be description of baseline models for comparison. Then, this subsection should be reduced and moved into experiment section and specify which related works are corresponding to each baseline model (RNN, LSTM, CNN1D). The authors do not need to describe the baseline models in details because those models are very common, well-known, and quite old and it would be enough just to specify related works to the baseline models. And datasets (3.1), experimental results (3.3.3) and some descriptions on hyper-parameters should also be moved into experiment section.

For the datasets used for experiments, the authors should provide where the dataset can be available or downloaded for reproducibility. Data-split (train-valid-test) should also be provided or specified which previous work was followed.

Subsection 3.3.1 (preprocessing) seems to be a novel point of this paper. However, some descriptions are vague and make it difficult to understand. The authors suggest two conversion approaches: gray image vs. RGB image. In the gray image approach, 256*256 pixel values are initialized to zero, and the Unicode value of each character is mapped into a coordinate <X,Y>, whose pixel value increases by one. Is this right? There are no description about pixel initialization and the amount of increase. And this coordinate mapping seems to be valid only for the gray image approach, not for the RGB image approach. However, the first paragraph (line 320-327) does not clarify this difference because it is not located within the description of the gray image approach, but located directly in the common upper-subsection 3.3.1. 

In the baseline approach (text-based), they remove all other characters except Korean and English characters in the preprocessing step. Is this same in the image-based approach? If not, why?

The authors add 128 to X-axis values of English characters to make it easy to identify them in the gray image approach. Who identifies them? Human? Machine has no difficulty in identifying the edge values.

Gray scale ranges from 0 (totally black) to 255 (totally white). In line 345-349, the authors increase the value of coordinate by 20 to increase the power of discrimination. But there is no description on how to determine this value. It quite depends on the length of target text data. The authors should suggest a reasonable method to determine that value based on the data statistics and how to clip too higher value than 256. By the way, Figure 9 seems not to be quite informative. Four sub-images seem to be just all black with the naked eye.

Gray image approach could be one of possible text-image conversion approaches, but it seems to have a severe disadvantage that it loses character sequence information, which would be very important in understanding the meaning of text. The authors should describe and analyze this characteristics with some experiments.

Is blank character just removed and ignored in the two conversion approaches? If yes, what is the reason?

In RGB image approach, the Unicode values of each character are mapped to R, G, and B values and the sequence of characters seems to remain in the converted image. But there are no description on the size of converted image. Is 256*256 size still used? 256*256 size means that the image can include up to 65,536 characters, which are quite long text. Since the target task data is SMS, which is quite short text in usual, most pixels of the image would be just padded. So, the authors need to suggest a reasonable size of image for SMS data.

In the view of multi-linguality, gray and RGB image approaches can convert all 2byte Unicode character to a pixel value of image. This is not limited to only English and Korean. To show this multi-linguality (i.e., language-independence)  of the suggest approaches, I recommend the authors would provide more experiments on various languages as well as English and Korean.

In subsection 3.3.2, the model’s details should be different for gray scale images and RGB images. And the last paragraph of this subsection describes the hyper-parameters used of experiments. That would be better to move the paragraph to the experiment section and represent it in a table.

The last activation could be softmax or sigmoid because the task is a kind of binary classification. However, the authors should clarify which one is used for the models. Some text (line 428) says sigmoid but some figure (eg, Fig 14) says softmax.

To prove the performance of the suggested method, the authors should compare to the most recent SOTA approaches. It seems that the authors selected RNN, LSTM and CNN 1D model as baselines but they are too simple and old approaches because they do not utilize any pre-trained word embeddings or any pre-trained language models. The authors should also compare their work to the recent SOTA performance of existing works used the same dataset (at least English SMS collection data).

The performance tables (Table 5, 7, 8) should be represented in the same (consistent) structure (column-based or row-based) or would be better to represented in just one table if possible. The scattered tables make it difficult to compare the performances.

The authors should specify how the performance value was obtained (averaged from how many runs)  because deep learning follows randomness.  And the performance (accuracy) values should be followed by standard deviations.

There is no analysis on the performance differences in each experiment. The baseline models show the accuracies for Korean are higher than the accuracies for English and between-accuracy for the Mixed data (only except for LSTM). On the contrary, gray scale image approach shows highest accuracy in English and RGB image approach shows highest accuracy in Mixed data. The authors should analyze and explain why this different performance trends occur. In addition, it is not generally understandable that the performance for mixed data is higher than performance for each separate language data. The authors should analyze and explain the reason.

Author Response

Dear , Reviewer 3,

Below is the answer to 'Comments and Suggestions for Authors', which you suggested and asked about our paper. And  For English writing, we assigned MDPI English editing service and got consulting on our overall style of sentence and vocabulary. Thank you for your nice comments. Please see the attachment.

Point 1:   English writing should be much improved by consulting with professional English editors. The citation reference position and style also should be corrected and improved. For example, “[1] According to …” in line 37-38, “Because of [14] and [15], “ in line 113, … should be clarified and corrected. 

Response 1: Thank you for your suggestion. For English writing, we assigned MDPI English editing service and got consulting on our overall style of sentence and vocabulary.  And for citation reference positions, we corrected several reference positions. We corrected “[1]According to…” to “In particular, a study [1]...” in(p2, lines 39), “Because of [14] and [15]...” to “Because of [15], [16]...” in (p3, lines 118) , “In [11], SVM and naive …” in (p3, lines 98), “... “vanishing gradient” problem [12]. ” in (p4,  line 102)

Point 2: Section 2 (related works) consists of two subsections: ‘text detection and classification’ and ‘visualization detection’. The section subsection is expected to describe existing works on conversion of text (string) into image, but any referred works do not deal with such conversion. I cannot distinguish the referred works in the second subsection from the referred works in the first subsection. And the referred works in the first subsection are also not well-classified and well-described. Similar(duplicate) description is repeated with different related works (1st and 2nd paragraphs in 2.1.1).

Response 2: We have reflected this comment by changing the title of subsection: ‘visualization detection’ into ‘spam detection using ML and DL’(p.5, line 176). And we modified 2.1.1(p3-4, lines 86-107)  to no paragraph separation, which had a similar description.  

Point 3: The authors should classify and describe recent existing works related to the suggested approach in the target task with which points are similar and which other points are different or unique. Most recent works in text mining field use pre-trained Language Model(PLM) like BERT and fine-tune it some output layers on top of PLM for each down-stream task like spam detection. The related works do not describe or compare the existing works on PLM-based spam detection to the suggested approach. 

Response 3: Thank you for providing these insights. We focus on multilingual processing analysis rather than performance difference. For this reason we decided to apply conventional models rather than recent models. But,we will be happy to consider the method you suggested and apply it in the future. Plus, we added “5. Discussion” section in (p16-17, lines 472-483) and mentioned your suggestion.

Point 4: The section 3 (proposed method) describes two different approaches with datasets. However, the first approach (in 3.2) must not be the suggested one by the authors. Instead, it would be description of baseline models for comparison. Then, this subsection should be reduced and moved into experiment section and specify which related works are corresponding to each baseline model (RNN, LSTM, CNN1D). The authors do not need to describe the baseline models in details because those models are very common, well-known, and quite old and it would be enough just to specify related works to the baseline models. And datasets (3.1), experimental results (3.3.3) and some descriptions on hyper-parameters should also be moved into experiment section.

Response 4: Thank you for your suggestion. Our explanation about the baseline model includes a detailed explanation of our model and when we write paper. And since we set the scope of our readers widely, we decided that including explanations about models frequently used for spam detection(RNN, LSTM, CNN 1D)  would be desirable for our readers to understand our papers. 

Experiment section can be divided into 3 parts <Datasets, Models, Results>. Among them, since the model section includes an overall explanation of our experiments, we keep the positions of section 3.1 Datasets and 3.3.3 Experimental results.  And for the description of hyper-parameters, we changed the position to Table 3(p.8, line 284), Table 4(p.9, line 297),Table 5(p.10, line 312), Table 8(p.15-16, line 439).

Point 5: For the datasets used for experiments, the authors should provide where the dataset can be available or downloaded for reproducibility. Data-split (train-valid-test) should also be provided or specified which previous work was followed.

Response 5: We have described sources of korean, english datasets in 3.1.Datasets (p.7-8, lines 251- 261). Each english dataset and emotional information in korean dataset can be downloaded in Kaggle and AIhub. Smishing dataset from Cyber Security BigData Center cannot be downloaded, as they were private dataset. Data-split has been reflected in Table 3(p.9, line 306), Table 4(p.9, line 320),Table 3(p.8, line 284), Table 4(p.9, line 297),Table 5(p.10, line 312), Table 8(p.15-16, line 439) by adding the validation_split value.

Point 6: Subsection 3.3.1 (preprocessing) seems to be a novel point of this paper. However, some descriptions are vague and make it difficult to understand. The authors suggest two conversion approaches: gray image vs. RGB image. In the gray image approach, 256*256 pixel values are initialized to zero, and the Unicode value of each character is mapped into a coordinate <X,Y>, whose pixel value increases by one. Is this right? There are no description about pixel initialization and the amount of increase. And this coordinate mapping seems to be valid only for the gray image approach, not for the RGB image approach. However, the first paragraph (line 320-327) does not clarify this difference because it is not located within the description of the gray image approach, but located directly in the common upper-subsection 3.3.1. 

Response 6: You have raised an important point. In the gray image approach, 256*256 pixel values are initialized to zero, and the Unicode value of each character is mapped into a coordinate (X,Y), whose pixel value increases by 200. We have rewritten  pixel initialization and the amount of increase  (p. 13, lines 386-391 ) to be more in line with your comments. 

And we changed the contents of the first paragraph of 3.3.1 (p.11-14, lines 322-388) to what the two conversion methods have in common, and revised the description of each method.

Point 7: In the baseline approach (text-based), they remove all other characters except Korean and English characters in the preprocessing step. Is this same in the image-based approach? If not, why?

Response 7: That's a good question. The image-based method removed space but did not remove special symbols. Special symbol removal is a preprocessing process. Unlike text-based character processing that creates meaning by combining special symbols, the image processing in our study does not require such a linguistic semantic interpretation. It is applied to avoid being restricted by the existing NLP processing process and unnecessary linguistic characteristics that recognize and limit special symbols as one character text.

Point 8: The authors add 128 to X-axis values of English characters to make it easy to identify them in the gray image approach. Who identifies them? Human? Machine has no difficulty in identifying the edge values.

Response 8: You have asked an interesting question. Machines identify X-axis values. When initially training the train data, features of the spam and ham message images were not identified, which was true for both machines and humans. Although you mentioned that the machine had no difficulty identifying the values ​​in the edge part, the results were different. We thought that it would show higher accuracy if we put the data clearly visible to the human eye into the train data, so we added various values, and added the generated images to make it easier for everyone to identify them.

Point 9: Gray scale ranges from 0 (totally black) to 255 (totally white). In line 345-349, the authors increase the value of coordinate by 20 to increase the power of discrimination. But there is no description on how to determine this value. It quite depends on the length of target text data. The authors should suggest a reasonable method to determine that value based on the data statistics and how to clip too higher value than 256. By the way, Figure 9 seems not to be quite informative. Four sub-images seem to be just all black with the naked eye.

Response 9: The reviewer was correct: we increased the value of coordinate by 200. The increase in pixel brightness has been divided by 10, 50, 100, 200 respectively. Based on the dataset statistics that more than 50 characters will not repeat, we set 200 as the max to prevent overflow. The pixel value 200, which the model detected best, was selected as the final increment.The pixel value is a dot code stamped per character, and 200 was selected because the recognition accuracy could be improved the most by giving the model such a difference in brightness, even if it is not distinguished by the human eye regardless of the data statistical point of view. As per the reviewer's suggestion, the existing figure 9 has been removed.

Point 10: Gray image approach could be one of possible text-image conversion approaches, but it seems to have a severe disadvantage that it loses character sequence information, which would be very important in understanding the meaning of text. The authors should describe and analyze this characteristics with some experiments.

Response 10: Thank you for your suggestion. As you commented in text based spam detection, character sequence information is important information. However, our methods visualization using Grayscale and RGB intentionally excluded character sequence information so that it can only consider individual unicode values of characters while creating images.  

Point 11: Is blank character just removed and ignored in the two conversion approaches? If yes, what is the reason?

Response 11: The reason for removing blank characters is that the semantic interpretation of Korean language is significantly different due to spacing. However, even though it is the same 2-byte character, blank characters are not used in cultures such as Chinese and Japanese, which use Chinese characters in the form of hieroglyphs. In using a method that does not process conditions according to the characteristics of each language, blank characters were removed because they required semantic interpretation rather than character recognition.

Point 12: In RGB image approach, the Unicode values of each character are mapped to R, G, and B values and the sequence of characters seems to remain in the converted image. But there are no description on the size of converted image. Is 256*256 size still used? 256*256 size means that the image can include up to 65,536 characters, which are quite long text. Since the target task data is SMS, which is quite short text in usual, most pixels of the image would be just padded. So, the authors need to suggest a reasonable size of image for SMS data.

Response 12: Some of the excessively long sentences (ex: over 1000) and meaningless sentences with less than 0 characters were removed from the dataset. And, among the remaining data sets, the size was specified as a criterion to which the longest sentence could be applied without being overly biased in data length.

Point 13: In the view of multi-linguality, gray and RGB image approaches can convert all 2byte Unicode character to a pixel value of image. This is not limited to only English and Korean. To show this multi-linguality (i.e., language-independence)  of the suggest approaches, I recommend the authors would provide more experiments on various languages as well as English and Korean

Response 13: We also considered other languages, but we decided that it is not applicable now due to the lack of time, and in 5.Discussion (p.16-17, lines 472-483), we presented the expected direction for applying to other studies in the future.

Point 14: In subsection 3.3.2, the model’s details should be different for gray scale images and RGB images. And the last paragraph of this subsection describes the hyper-parameters used of experiments. That would be better to move the paragraph to the experiment section and represent it in a table.

Response 14: Since the purpose of this method was to see the difference in accuracy depending on the image creation method when image-based spam detection was performed with CNN 2D, the rest of the model details except for the image creation method were tried to proceed under the same conditions as much as possible.

As mentioned above, the main body is divided into three parts: <dataset, model, and experimental results>. Among them, the model section can be regarded as the experiment section, including the overall description of the experiment. Therefore, the description of hyperparameters was added as Table 8(p.15-16, line 439) of the experiment section according to the reviewer's suggestion.

Point 15: The last activation could be softmax or sigmoid because the task is a kind of binary classification. However, the authors should clarify which one is used for the models. Some text (line 428) says sigmoid but some figure (eg, Fig 14) says softmax..

Response 15:  We agree with you and have incorporated your comments by editing Fig 12 (p.15, line 438) softmax to sigmoid. 

Point 16: To prove the performance of the suggested method, the authors should compare to the most recent SOTA approaches. It seems that the authors selected RNN, LSTM and CNN 1D model as baselines but they are too simple and old approaches because they do not utilize any pre-trained word embeddings or any pre-trained language models. The authors should also compare their work to the recent SOTA performance of existing works used the same dataset (at least English SMS collection data).

Response 16: As mentioned in response 3, the reason why the latest model was not applied is that good results were also obtained with the existing model. As you suggested, we will research it later. Related contents were added to 5.Discussion (p.16-17, lines 472-483). Thank you for your suggestion.

Point 17: The performance tables (Table 5, 7, 8) should be represented in the same (consistent) structure (column-based or row-based) or would be better to represented in just one table if possible. The scattered tables make it difficult to compare the performances.

Response 17: Thank you for your suggestion. Table 6 (p.10, lines 318) shows performance of text based models accuracy and table 7,8 which is now deleted shows performance of image based models accuracy.Agreed with you, we combined former table 7,8 and created table 9 (p.16, lines 446).

Point 18: The authors should specify how the performance value was obtained (averaged from how many runs)  because deep learning follows randomness.  And the performance (accuracy) values should be followed by standard deviations.

Response 18: Thank you for your suggestion. Our averaged number of how many runs are mentioned in Table 3(p.8, line 284), Table 4(p.9, line 297),Table 5(p.10, line 312), Table 8(p.15-16, line 439) epochs. Also we set early stopping conditions so that models automatically stop when the same result is repeated.

Point 19: There is no analysis on the performance differences in each experiment. The baseline models show the accuracies for Korean are higher than the accuracies for English and between-accuracy for the Mixed data (only except for LSTM). On the contrary, gray scale image approach shows highest accuracy in English and RGB image approach shows highest accuracy in Mixed data. The authors should analyze and explain why this different performance trends occur. In addition, it is not generally understandable that the performance for mixed data is higher than performance for each separate language data. The authors should analyze and explain the reason.

Response 19: Thank you for your suggestion. Our study focuses on multilingual processing analysis rather than performance difference. Our study results show over 99% of accuracy in all cases ( English / Korean / Mixed). Through these results, we discovered that applying the visualization process also worked well in mixed datasets. And as the difference in the accuracy is extremely fine, we do not proceed with the analysis of the result related to this. Since various styles of multilingual languages also produced significant results from our deep learning models, this can be a major contribution of our study.

Thank you.

Round 2

Reviewer 1 Report

The authors have satisfactorily addressed my comments and I may recommend the manuscript for publication